**Data Availability Statement:** Due to legal restrictions (under New York Public Health Law Article 21, Title III) and the confidential nature of HIV surveillance data in New York, public health

# Service utilization and HIV outcomes among transgender women receiving Ryan White Part A services in New York City

**Jacinthe A. Thomas**⊙*, **Mary K. Irvine, Qiang Xia, Graham A. Harriman**

New York City Department of Health and Mental Hygiene, Queens, New York, United States of America

* jthomas1@health.nyc.gov

## Abstract

### Background

Prior research has found evidence of gender disparities in U.S. HIV healthcare access and outcomes. In order to assess potential disparities in our client population, we compared demographics, service needs, service utilization, and HIV care continuum outcomes between transgender women, cisgender women, and cisgender men receiving New York City (NYC) Ryan White Part A (RWPA) services.

### Methods

The analysis included HIV-positive clients with an intake assessment between January 2016 and December 2017 in an NYC RWPA services program. We examined four service need areas: food and nutrition, harm reduction, mental health, and housing. Among clients with the documented need, we ascertained whether they received RWPA services targeting that need. To compare HIV outcomes between groups, we applied five metrics: engagement in care, consistent engagement in care, antiretroviral therapy (ART) use, point-in-time viral suppression, and durable viral suppression.

### Results

All four service needs were more prevalent among transgender women (N = 455) than among cisgender clients. Except in the area of food and nutrition services, timely (12-month) receipt of RWPA services to meet a specific assessed need was not significantly more or less common in any one of the three client groups examined. Compared to cisgender women and cisgender men, a lower proportion of transgender women were durably virally suppressed (39% versus 52% or 50%, respectively, p-value < 0.001).

### Conclusions

Compared with cisgender women and cisgender men, transgender women more often presented with basic (food/housing) and behavioral-health service needs. In all three groups (with no consistent between-group differences), assessed needs were not typically met with the directly corresponding RWPA service category. Targeting those needs with RWPA

authorities in New York City cannot release individual-level data on reported HIV cases for purposes other than ensuring appropriate HIV care. This restriction applies even to de-identified patient-level datasets. However, NYC DOHMH staff are available to assist external researchers who may have further specific data questions or uses. An email can be sent to hivreport@health.nyc.gov with questions or requests for additional information, which will be answered promptly by NYC DOHMH staff.

**Funding:** This work was supported through a grant from the Health Resources and Services Administration [H89HA00015] to the New York City Department of Health and Mental Hygiene. The funders had no role in study design, data collection and analysis, decision to publish, or preparation of the manuscript.

**Competing interests:** The authors have declared that no competing interests exist.

outreach and services may support the National HIV/AIDS Strategy 2020 goal of reducing health disparities, and specifically the objective of increasing (to ≥90%) the percentage of transgender women in HIV medical care who are virally suppressed.

## Introduction

The federal Ryan White HIV/AIDS Program (RWHAP) has been a critical safety net for people living with HIV (PLWH) for the past thirty years [1]. The program funds medical and supportive services for individuals without sufficient alternative resources to manage HIV disease. RWHAP services are designed to support engagement in HIV medical care and address psychosocial or structural barriers to viral suppression (reduction of HIV-1 viral load [VL] in plasma to levels below the detection limit of common HIV RNA tests used by healthcare providers). More than half of all U.S. PLWH are RWHAP clients [2]. In 2018, 87% of RWHAP clients were virally suppressed, exceeding the national viral suppression average of 63% [3]. However, while RWHAP services appear to have a positive effect on treatment outcomes [4, 5], significant inequities exist among clients [6], particularly by gender identity. Overall, in 2018, transgender women in RWHAP experienced lower retention in care and viral suppression (78% retention and 81% suppression) than cisgender women in RWHAP (83% retention and 87% suppression); these outcome disparities were observed in each housing status category (unstably housed, temporarily housed, and stably housed) [3]. Prior research in the U.S. has shown that transgender women, compared with other PLWH, face barriers to engagement in care and antiretroviral treatment (ART) adherence due to negative experiences with healthcare providers [7, 8].

Multiple New York City (NYC) reports have also shown evidence of gender disparities in HIV outcomes. A citywide study based on HIV surveillance data found that transgender women had lower viral suppression than men who have sex with men (MSM) in the first 12 months after HIV diagnosis [9]. Similarly, a more recent analysis highlighted significant differences in viral suppression among transgender women, cisgender women, and MSM receiving Ryan White Part A (RWPA) services [10]. In addition, NYC Health Department data from a 2014 client satisfaction survey showed that transgender women respondents were less likely to be "very satisfied" with their RWPA services or the way they were treated overall, compared to cisgender women respondents [10]. Transgender women surveyed were also significantly less likely than cisgender women to "agree" or "strongly agree" with statements indicating that they were treated with respect and that all staff were nice to them. In the same survey, qualitative feedback from transgender women clients highlighted unmet service needs for food and housing [10].

While very few studies have examined care continuum outcomes among transgender women compared with cisgender women and men enrolled in RWHAP [11], no published studies have compared transgender and cisgender RWHAP clients on both service utilization and care continuum outcomes. For the current analysis, we examined differences between transgender women, cisgender women, and cisgender men in NYC RWPA with regard to: 1) demographics and service utilization; 2) four areas of service need: food and nutrition, harm reduction, mental health, and housing services, and the extent to which those needs were met with directly related NYC RWPA supportive services; and 3) HIV outcomes along the care continuum, including engagement in care, ART use, point-in-time viral suppression and durable viral suppression. Assessing and understanding unmet service needs and HIV outcome disparities experienced by transgender NYC RWPA clients can inform the development of

strategies to intervene and ultimately to advance gender equity in HIV care continuum outcomes.

## Materials and methods

### Inclusion criteria

The analysis included HIV-positive clients with at least one complete NYC RWPA assessment, from January 2016 to December 2017, in a contract for medical or non-medical case management, mental health or supportive counseling, food and nutrition, harm reduction, housing, legal, and/or health education/risk reduction services. These NYC RWPA service categories were selected because they focus on HIV-diagnosed individuals, share a core set of assessment questions relevant to service needs, and follow clients beyond the initial linkage to care. New York State (NYS) eligible metropolitan area (EMA) RWPA clients in these service categories are assessed upon enrollment in order to identify specific needs for services and devise a plan to address them. They are also reassessed approximately every six months thereafter while they remain enrolled, allowing for the identification of persistent and emerging needs, to guide client-centered service planning and coordination.

### Data sources

Data on enrollments, demographics, and receipt of RWPA services, as well as assessments, were drawn from the Electronic System for HIV/AIDS Reporting and Evaluation (eSHARE), a secure, Web-based reporting system for HIV services contracts. NYC client-level RWPA data are routinely matched against data from the NYC HIV Surveillance Registry (the "Registry") using a deterministic matching process, which has been described elsewhere [12]. For the current analysis, provider-reported programmatic data in eSHARE were merged with the Registry to link individual RWPA client records with complete NYC laboratory (VL and CD4) test records. Electronically captured, named reporting of all HIV-related laboratory tests, including positive diagnostic tests and viral nucleotide sequences, has been mandatory in NYS since 2005 [13].

### Ethics

Participant consent did not apply for this retrospective analysis, which utilized secondary data reported to the NYC Health Department as required under NYS law or under contractual agreements with RWPA-funded agencies. The HIV surveillance and RWPA programmatic data sets were fully de-identified prior to analysis. Only authorized Health Department analysts trained in HIV confidentiality and data security protocols have access to these data sets. This analysis met the definition of public health surveillance and was designated as not human subjects research by the NYC Health Department institutional review board (IRB).

### Demographic measures

Using information from eSHARE, we defined transgender women as clients having either self-identified gender reported as "transgender woman or girl" or self-identified gender reported as "woman or girl" and sex assigned at birth reported as "male." Cisgender women clients were defined as having self-identified gender reported as "woman or girl" and sex assigned at birth reported as "female." Similarly, we used self-identified gender (reported as "man or boy") and sex assigned at birth (reported as "male") to classify cisgender men. Transgender men and non-binary individuals were not included as separate groups because of their small numbers (<35 and <5, respectively) in the client cohort available for analysis. eSHARE also captures

age, race and ethnicity, country of birth, educational attainment, employment status, and primary language. Race and ethnicity were combined into a single race/ethnicity variable, for which clients who identified as having Hispanic ethnicity were categorized as "Hispanic/Latinx" regardless of reported race, and non-Hispanic/Latinx clients reporting only one race (Black or White) were categorized as that race, while a relatively small number of non-Hispanic/Latinx clients reporting Asian race, more than one race, or "Other" race were included in a combined "Asian/Other/Mixed race" category, and non-Hispanic/Latinx clients with entirely missing race data were included in the "Unknown" category. We combined race and ethnicity into a single variable in part because data on race were disproportionately missing among clients for whom Hispanic ethnicity was reported. Educational attainment was categorized as "below high school" and "at or above high school/general educational development (GED)" (for completion of high school or any amount of higher education). Employment status, which is based on five mutually exclusive levels, was categorized as "employed" (for full-time or part-time employment); "unemployed" (for unemployed or unpaid volunteer/peer worker status); and "out of workforce" (for student, retired or homemaker status).

## Service need and utilization measures

RWPA clients can be enrolled in more than one program or service category at a time. To determine service category utilization, we checked for receipt of any service between January 1, 2016 and December 31, 2017 in the service categories included in this analysis (i.e. medical or non-medical case management, mental health or supportive counseling, food and nutrition, harm reduction, housing, legal and/or health education/risk reduction services). Using data from assessments that were completed between January 1, 2016 and December 31, 2017, we defined need according to the following criteria. Food and nutrition need was defined as food insufficiency (FI) or income ≤130% of Federal Poverty Level (FPL), based on the eligibility cutoff in NYS for the Supplemental Nutritional Assistance Program (SNAP) [14]. FI status was assessed based on responses to the following questions, collected in only some of the service categories included in the analysis: 1) "In the past three months, how often has it happened that there was not enough money for food in the household?" 2) "Which of the following best describes your situation in terms of food you eat?" and 3) "In the last 30 days, did you go a whole day without anything at all to eat (because you did not have adequate access to food)?" Clients were classified as having FI if they reported (1) "once in a while," "fairly often," or "very often" not having enough money for food in the past 3 months; (2) "sometimes" or "often" not having enough to eat; or (3) going for a whole day without anything at all to eat in the past 30 days. Harm reduction need was defined as use of cocaine/crack, heroin, crystal methamphetamine, or prescription drugs to get high in the past three months. Mental health need was defined as a mental component summary score of ≤37.0 on the 12-item Short Form survey [15]. Housing need was defined as unstable housing, which includes homelessness and transitional/temporary housing situations.

We chose to analyze these four areas of need because they each have a parallel RWPA service category that is funded within the NYS EMA. While some other service categories funded locally are designed to respond to a wider range of needs or barriers to care/treatment engagement (e.g., non-medical or medical case management), others (e.g., oral healthcare) address needs that have not been directly assessed in eSHARE or could be considered universal. Among transgender women, cisgender women and cisgender men with evidence of a need at intake assessment between January 1, 2016 and December 31, 2017, we ascertained whether they received NYC RWPA services targeting that need within 12 months after the first assessment indicating that need.

### HIV outcome measures

HIV outcomes were derived from HIV-related laboratory test information from the Registry. We used five metrics: 1) engagement in care, defined as having at least one HIV-related (VL or CD4) laboratory test within 12 months after the last service received between January 1, 2016 and December 31, 2017 (observation period); 2) consistent engagement in care, defined as having at least two HIV-related (VL or CD4) laboratory tests, at least two months apart, within 12 months after the last service received in the observation period; 3) ART use, defined as having a current ART prescription in the observation period; 4) point-in-time viral suppression, defined as having a value <200 copies/mL on the latest VL test result within 12 months after the last service received in the observation period; and 5) durable viral suppression, defined as having at least two VL test results at least 2 months apart within 12 months after the last service received in the observation period, and having values <200 copies/mL on all VL results in that 12-month follow-up period. Clients without a VL test in that timeframe were classified as virally unsuppressed.

### Data analysis

All analyses were performed in SAS version 9.4. Overall differences in service utilization and HIV outcomes between the three groups were first assessed using the χ2 test or the Fisher's exact test (where appropriate). Where initial tests showed any significant gender disparity, post-hoc tests were used for pairwise comparisons. Poisson regression models were used to estimate adjusted prevalence ratios of HIV care continuum outcomes for transgender women and cisgender men compared to cisgender women, controlling for age, race/ethnicity, and country of birth [16]. We selected these variables as covariates based on a causal model using a directed acyclic graph [17].

## Results

Clients eligible for the overall analysis included 455 transgender women (3%), 4,906 cisgender women (33%), and 9,699 cisgender men (64%). The demographic characteristics of the three groups are listed in Table 1. In all three gender groups, most clients were Black or Hispanic, U.S.-born, primarily English speaking, and unemployed. There were differences between the three groups on all demographic characteristics. Compared to cisgender women and cisgender men, transgender women were more likely to be younger (21% versus 7% and 10%) and Hispanic/Latinx (43% versus 32% and 38%). A higher proportion of transgender women were unemployed (75%) and reported Spanish as their primary language (24%), compared to the two other groups. Transgender women were more likely to be born outside of the U.S. (28%) and to have education at or above the high school/GED level (60%) compared to cisgender women (24% and 50%), but cisgender men were more likely than transgender women to have that higher level of educational attainment (66%).

Transgender women were significantly more likely to receive harm reduction services (24% versus 14% for both cisgender women and cisgender men) and housing assistance (23% versus 9% and 11% for cisgender women and cisgender men, respectively) (Table 2). However, compared to cisgender men, a significantly lower proportion of transgender women received food and nutrition services (23% versus 31%). Transgender women were also less likely to receive legal services (17% versus 25% and 24% for cisgender women and cisgender men, respectively) (Table 2).

Compared to cisgender women and cisgender men, significantly higher proportions of transgender women had an apparent need for support in the areas of housing (52% versus 24% and 35%, respectively), harm reduction (23% versus 12% and18%, respectively), and food

**Table 1. Demographic characteristics of HIV-positive Ryan White Part A clients served and assessed between January 1, 2016 and December 31, 2017.**

| Characteristics | Transgender Women (N = 455) | | Cisgender Women (N = 4906) | | Cisgender Men (N = 9699) | |
|---|---|---|---|---|---|---|
| | N | % | N | % | N | % |
| **Age group (years)** | | | | | | |
| Under 30 | 94 | 21 | 355 | 7 | 975 | 10 |
| 30-49 | 255 | 56 | 1654 | 34 | 3735 | 39 |
| 50 or older | 106 | 23 | 2897 | 59 | 4989 | 51 |
| **Race and ethnicity** | | | | | | |
| Black | 217 | 48 | 2992 | 61 | 4574 | 47 |
| Hispanic/Latinx | 197 | 43 | 1563 | 32 | 3653 | 38 |
| White | 21 | 5 | 220 | 4 | 1073 | 11 |
| Asian/Other/Mixed race | 18 | 4 | 97 | 2 | 352 | 4 |
| Unknown | 2 | <1 | 34 | <1 | 47 | <1 |
| **Country of birth** | | | | | | |
| U.S./U.S. territories | 321 | 71 | 3604 | 73 | 6666 | 69 |
| Outside of the U.S. | 126 | 28 | 1187 | 24 | 2766 | 29 |
| Unknown | 8 | 2 | 115 | 2 | 267 | 3 |
| **Educational attainment** | | | | | | |
| Below high school | 165 | 36 | 2244 | 46 | 2859 | 29 |
| At or above high school/GED | 271 | 60 | 2462 | 50 | 6406 | 66 |
| Unknown | 19 | 4 | 200 | 4 | 434 | 4 |
| **Employment status** | | | | | | |
| Employed | 48 | 11 | 564 | 12 | 1596 | 16 |
| Unemployed | 340 | 75 | 3468 | 71 | 6650 | 69 |
| Out of workforce | 57 | 13 | 792 | 16 | 1248 | 13 |
| Unknown | 10 | 2 | 82 | 2 | 205 | 2 |
| **Primary language** | | | | | | |
| English | 328 | 72 | 3794 | 77 | 7250 | 75 |
| Spanish | 111 | 24 | 784 | 16 | 1972 | 20 |
| Other | 11 | 2 | 314 | 6 | 432 | 4 |
| Unknown | 5 | 1 | 14 | <1 | 45 | <1 |

GED, general educational development; HIV, human immunodeficiency virus.

Percentages may not add to 100% within a client subgroup, because of rounding.

and nutrition (95% versus 92% and 88%, respectively) (Table 3). Transgender women also had a higher apparent need for mental health services, as compared with cisgender men (24% versus 20%). A smaller proportion of transgender women with food and nutrition service needs received those services in the following 12 months (23%), compared to cisgender women (28%) and cisgender men (33%) with the same need (Table 4). Otherwise, timely (12-month) receipt of RWPA services to meet a specific assessed need was not significantly more or less common in any one of the three client groups examined.

Of the five care continuum metrics that we examined (Tables 5 and 6), only the durable viral suppression measure was substantially and significantly different between transgender women and the other two groups. There were significant overall gender differences in the unadjusted analyses for consistent engagement in care and ART prescription status, but pair-wise comparisons showed no significant disadvantage for transgender women relative to cisgender women or to cisgender men (Table 5). Compared to cisgender women and cisgender men, a lower proportion of transgender women clients had durable viral suppression (39%

**Table 2. Ryan White Part A service category utilization by gender, 2016-2017.**

| Service Category | TW (N = 455) | CW (N = 4906) | CM (N = 9699) | Overall | TW vs. CW | | TW vs. CM | | CW vs. CM | |
|---|---|---|---|---|---|---|---|---|---|---|
| | N (%) | N (%) | N (%) | P-value* | Chi-square | P-value | Chi-square | P-value | Chi-square | P-value |
| **Case Management** | 194 (43) | 2021 (41) | 3983 (41) | 0.7995 | — | — | — | — | — | — |
| **Harm Reduction** | 111 (24) | 704 (14) | 1400 (14) | **<.001** | 35.08 | **<.001** | 37.10 | **<.001** | 0.02 | 0.8905 |
| **Housing** | 106 (23) | 423 (9) | 1084 (11) | **<.001** | 105.15 | **<.001** | 66.97 | **<.001** | 22.60 | **<.001** |
| **Food and Nutrition** | 104 (23) | 1324 (27) | 2998 (31) | **<.001** | 3.46 | 0.0628 | 11.91 | **<.001** | 23.56 | **<.001** |
| **Mental Health** | 94 (21) | 1122 (23) | 1753 (18) | **<.001** | 1.13 | 0.2873 | 2.01 | 0.1566 | 47.96 | **<.001** |
| **Legal** | 77 (17) | 1234 (25) | 2361 (24) | **<.001** | 13.78 | **<.001** | 11.9 | **<.001** | 1.16 | 0.2822 |
| **Health Education/Risk Reduction** | 44 (10) | 429 (9) | 844 (9) | 0.7746 | — | — | — | — | — | — |

TW, transgender women; CW, cisgender women; CM, cisgender men.

*P-value is based on the chi-square/Fisher's exact test as applicable.

Dashes signify that pairwise comparisons were not conducted when no significant main effect was found.

Significant p-values are bolded.

versus 52% or 50%, respectively, p-value < 0.001). The adjusted prevalence ratio (aPR) shows that transgender women were less likely to have durable viral suppression than cisgender women (aPR: 0.80, 95% CI: 0.69-0.94), after controlling for age, race/ethnicity, and country of birth (Table 6).

## Discussion

Our analysis compared demographics, service needs and utilization, and HIV care continuum outcomes for transgender women, cisgender women, and cisgender men enrolled in NYC RWPA programs. Transgender women, compared to cisgender women and cisgender men in our sample, tended to be younger and more often Hispanic/Latinx, primarily Spanish-speaking, and unemployed.

We found significant differences in service category utilization for the three groups, with a lower use of food and nutrition services among transgender women (relative to cisgender men) and a lower use of legal services (relative to cisgender men and to cisgender women). Transgender women in NYC RWPA were more likely than cisgender women and cisgender men to have a documented need for food and nutrition, harm reduction, or housing assistance. Furthermore, compared to cisgender men, transgender women were more likely to have documented need for mental health services. The high prevalence of basic and

**Table 3. Ryan White Part A service category need by gender.**

| Service Area | TW (N = 455) | CW (N = 4906) | CM (N = 9699) | Overall | TW vs. CW | | TW vs. CM | | CW vs. CM | |
|---|---|---|---|---|---|---|---|---|---|---|
| | Had Need | Had Need | Had Need | P-value* | Chi-square | P-value | Chi-square | P-value | Chi-square | P-value |
| | N (%) | N (%) | N (%) | | | | | | | |
| **Food/Nutrition** | 431 (95) | 4514 (92) | 8528 (88) | **<.001** | 6.04 | **0.014** | 40.63 | **<.001** | 64.68 | **<.001** |
| **Harm Reduction** | 105 (23) | 600 (12) | 1737 (18) | **<.001** | 45.87 | **<.001** | 8.24 | **0.0041** | 75.17 | **<.001** |
| **Mental Health** | 107 (24) | 1100 (22) | 1922 (20) | **<.001** | 0.29 | 0.5906 | 3.87 | **0.049** | 13.59 | **<.001** |
| **Housing** | 238 (52) | 1167 (24) | 3419 (35) | **<.001** | 233.7 | **<.001** | 71.01 | **<.001** | 183.66 | **<.001** |

TW, transgender women; CW, cisgender women; CM, cisgender men.

*P-value is based on the chi-square/Fisher's exact test as applicable.

Significant p-values are bolded.

**Table 4. Ryan White Part A service category utilization by gender among those with assessed need – within 12 months after the assessment indicating the need.**

| Service Area | TW | | CW | | CM | | Overall | TW vs. CW | | TW vs. CM | | CW vs. CM | |
|---|---|---|---|---|---|---|---|---|---|---|---|---|---|
| | Had need | Had service | Had Need | Had service | Had Need | Had service | P-value* | Chi-square | P-value | Chi-square | P-value | Chi-square | P-value |
| | N(%) | N (%) | N (%) | N (%) | N (%) | N (%) | | | | | | | |
| Food/Nutrition | 431 (95) | 99 (23) | 4514 (92) | 1246 (28) | 8528 (88) | 2785 (33) | <.001 | 4.04 | **0.0445** | 15.43 | **<.001** | 34.36 | **<.001** |
| Harm Reduction | 105 (23) | 71 (68) | 600 (12) | 348 (58) | 1737 (18) | 1080 (62) | 0.0788 | — | — | — | — | — | — |
| Mental Health | 107 (24) | 47 (44) | 1100 (22) | 506 (46) | 1922 (20) | 811 (42) | 0.1273 | — | — | — | — | — | — |
| Housing | 238 (52) | 150 (63) | 1167 (24) | 671 (58) | 3419 (35) | 2045 (60) | 0.1931 | — | — | — | — | — | — |

TW, transgender women; CW, cisgender women; CM, cisgender men.

*P-value is based on the chi-square/Fisher's exact test as applicable.

Had service: Among those with a need.

Dashes signify that pairwise comparisons were not conducted when no significant main effect was found.

Significant p-values are bolded.

behavioral-health needs among transgender women may correspond to structural inequities and barriers faced by this population, including gender-related stigma and discrimination and racial/ethnic and linguistic discrimination, which could diminish economic opportunity and negatively impact health [18]. Transgender women, however, were just as likely as cisgender women and cisgender men to receive RWPA services to address their harm reduction, housing, and/or mental health needs.

The observed lower use of food and nutrition services among transgender women with assessed need may have to do in part with our definition of need for food and nutrition services, which for many clients was based on the FPL measure alone, since FI questions are not collected from all service categories in NYC RWPA. Because transgender women tend to experience high rates of poverty [18, 19], they may have been classified as having a need for food and nutrition based on their income even if they were not experiencing FI. In addition, some food and nutrition services, especially home delivery of meals, are more focused on people with disabilities or chronic illnesses other than HIV, including PLWH who are older or homebound. Relative to the overall NYC RWPA client population, transgender women clients are younger and might not be as frequently perceived as needing home delivery of meals.

Our results on service needs are consistent with those of Mizuno et al.'s analysis of data from the Medical Monitoring Project (MMP) [20], showing a higher proportion of

**Table 5. HIV outcomes among Ryan White Part A clients by gender, based on HIV surveillance.**

| Service Category | TW (N = 455) | CW (N = 4906) | CM (N = 9699) | Overall | TW vs. CW | | TW vs. CM | | CW vs. CM | |
|---|---|---|---|---|---|---|---|---|---|---|
| | N (%) | N (%) | N (%) | P-value* | Chi-square | P-value | Chi-square | P-value | Chi-square | P-value |
| Engagement in care | 453 (99.6) | 4892 (99.7) | 9650 (99.5) | 0.16 | — | — | — | — | — | — |
| Consistent engagement in care | 447 (98.2) | 4856 (99.0) | 9548 (98.4) | **0.0253** | 1.36 | 0.2443 | 0.10 | 0.7490 | 7.96 | **0.0048** |
| ART prescription | 420 (92.3) | 4484 (91.4) | 8803 (90.8) | **0.0404** | 2.47 | 0.1159 | 0.74 | 0.3892 | 5.20 | **0.0226** |
| Viral suppression | 347 (73.3) | 3956 (80.6) | 7781 (80.2) | 0.0811 | — | — | — | — | — | — |
| Durable viral suppression | 175 (38.5) | 2562 (52.2) | 4830 (49.8) | <.001 | 25.26 | **<.001** | 18.43 | **<.001** | 7.77 | **0.0053** |

TW, transgender women; CW, cisgender women; CM, cisgender men; ART, antiretroviral therapy; HIV, human immunodeficiency virus.

*P-value is based on the chi-square/Fisher's exact test as applicable.

Dashes signify that pairwise comparisons were not conducted when no significant main effect was found.

Significant p-values are bolded.

**Table 6. Adjusted prevalence ratios of HIV care outcomes among Ryan White Part A clients, based on HIV surveillance.**

|  | Cisgender women Reference | Cisgender men (95% CI) | Transgender women (95% CI) |
|---|---|---|---|
| **Engagement in care** | 1.00 | 1.00 (0.96, 1.03) | 1.00 (0.91, 1.10) |
| **Consistent engagement in care** | 1.00 | 1.00 (0.96, 1.03) | 0.99 (0.90, 1.10) |
| **ART prescription** | 1.00 | 0.99 (0.95, 1.03) | 0.99 (0.89, 1.09) |
| **Viral suppression** | 1.00 | 0.98 (0.94, 1.02) | 0.97 (0.87, 1.09) |
| **Durable viral suppression** | 1.00 | 0.92 (0.88, 0.97) | 0.80 (0.69, 0.94) |

ART, antiretroviral therapy; CI, confidence interval; HIV, human immunodeficiency virus.

Prevalence ratios were adjusted for age, race/ethnicity, and country of birth.

transgender women having a need for supportive services, as compared to non-transgender persons. These findings suggest the urgency of closing gaps in services for transgender women living with HIV, since unmet basic/material and behavioral health-related needs have been repeatedly demonstrated to negatively impact HIV treatment outcomes [21–23].

In bivariate analyses, we found overall significant differences between transgender women and cisgender women or cisgender men with regards to consistent engagement in care. Although a slightly higher proportion of transgender women were prescribed ART, this favorable outcome was not translated into an advantage for short-term or durable viral suppression. Transgender women were less likely to be virally suppressed or durably virally suppressed, although only the latter result reached statistical significance. This is consistent with previous findings showing gender disparities in viral suppression [11, 20, 24]. These disparities may be due in part to the expression of gender-related bias in negative interactions with providers, stigma, and mistreatment in service settings [7, 8, 10]. Our findings may also reflect the particular challenge of maintaining daily adherence to ART over time in the face of persistent barriers to treatment, as compared with achieving shorter-term or lower-threshold outcomes like twice-yearly medical visits, ART initiation, or even point-in-time viral suppression [25].

Existing research highlights some potential ways in which service utilization may be related to HIV care continuum outcomes. For example, a previous study among NYC RWPA clients enrolled in a medical case management program known as Care Coordination has shown that persistent challenges such as low mental health functioning, hard drug use, or unstable housing function as barriers to desired outcomes, but the resolution of these barriers tends to be associated with greater improvement in care engagement and/or viral suppression [26]. This finding reinforces the importance of addressing unmet psychosocial and structural needs as a way to optimize viral suppression. In addition, given the immense influence of stable housing on HIV outcomes [27, 28] and the disproportionate housing instability among transgender women in NYC RWPA, efforts to better meet housing needs will be important to advancing gender equity in viral suppression.

Our analysis has several limitations. First, ascertainment of gender identity from program reporting can be subject to error; however, we trained and provided guidance to NYC RWPA service providers on how to collect these data via client self-report. Second, the definitions that we used for the four areas of need are limited in sensitivity and specificity. The standardized assessment tools used for routine RWPA reporting are not designed to function as comprehensive screening tools, particularly with regard to complex conditions such as mental health or substance use disorders. Third, we only account for receipt of services during the 12-month observation period after the assessment indicating the need, so some clients who did not receive the targeted RWPA service during the observation period may still have had that need met at a later point. In addition, given that RWPA services are used in NYC to fill gaps in

services available through other payers, many of the needs identified by RWPA providers and clients may ultimately be addressed through linkages to (or independent client utilization of) non-RWPA support services, such as those available through Medicaid, SNAP, other Parts (B, C or D) of the RWHAP and/or Housing Opportunities for Persons with AIDS (HOPWA), including the NYC HIV/AIDS Services Administration (HASA). However, even in prior analyses based on integrated RWPA and HOPWA housing service data, which substantially improved ascertainment of housing assistance, assessed housing need was still not consistently met with a RWPA or HOPWA housing service among transgender women [29].

## Conclusions

In all four need areas we could measure across multiple programs, transgender women in NYC RWPA more often presented with service needs than cisgender clients, although service need for mental health was not significantly different when compared to cisgender women, specifically. For all three groups of clients, we found that assessed needs were not consistently met with the corresponding RWPA support service within a 12-month period, and there were no consistent differences between groups in rates of receipt of services, though transgender women were significantly less likely to receive food/nutrition services to address apparent need in that area. Future studies should pursue greater integration of services data across major public payers and data sources (e.g., Medicaid, other parts of Ryan White, SNAP), to better isolate unmet service needs. Simultaneous efforts to strengthen outreach, promote gender-affirming service delivery and increase engagement of transgender women in RWPA programs addressing basic survival and behavioral health needs can support the National HIV/AIDS Strategy 2020 goal of reducing health disparities, and specifically the objective of increasing (to ≥90%) the percentage of transgender women in HIV medical care who are virally suppressed [30].

## Acknowledgments

The authors are indebted to: Kristina Rodriguez, Melanie Lawrence and Scott Spiegler for their contribution through their involvement with the Care and Treatment Race to Justice Transgender Women of Color Workgroup; Anisha Gandhi, Matthew Feldman and Kent Sepkowitz for their helpful critiques of initial drafts; and NYC Ryan White Part A service providers for their dedication to the delivery of services.

## Author Contributions

**Conceptualization:** Jacinthe A. Thomas, Mary K. Irvine.

**Data curation:** Jacinthe A. Thomas.

**Formal analysis:** Jacinthe A. Thomas.

**Methodology:** Jacinthe A. Thomas, Mary K. Irvine, Qiang Xia.

**Project administration:** Graham A. Harriman.

**Supervision:** Mary K. Irvine.

**Writing – original draft:** Jacinthe A. Thomas.

**Writing – review & editing:** Mary K. Irvine, Qiang Xia, Graham A. Harriman.

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
