## [Decision Letter · Decision Letter 0]

10 Dec 2020

PONE-D-20-32483

Service utilization and HIV outcomes among transgender women receiving Ryan White Part A services in New York City

PLOS ONE

Dear Dr. Thomas,

Thank you for submitting your manuscript to PLOS ONE. After careful consideration, we feel that it has merit but does not fully meet PLOS ONE’s publication criteria as it currently stands. Therefore, we invite you to submit a revised version of the manuscript that addresses the points raised during the review process.

My own comments are as Reviewer 2 below, however, I agree with the other reviewer's comments as well. Please consider each carefully and incorporate as you see fit. I look forward to receiving your revisions in due course.

We look forward to receiving your revised manuscript.

Kind regards,

Ethan Morgan

Academic Editor

PLOS ONE

Journal Requirements:

2.Please provide additional details regarding participant consent. In the ethics statement in the Methods and online submission information, please ensure that you have specified (1) whether consent was informed and (2) what type you obtained (for instance, written or verbal, and if verbal, how it was documented and witnessed). If your study included minors, state whether you obtained consent from parents or guardians. If the need for consent was waived by the ethics committee, please include this information.

3.We note that you have indicated that data from this study are available upon request. PLOS only allows data to be available upon request if there are legal or ethical restrictions on sharing data publicly. For information on unacceptable data access restrictions, please see http://journals.plos.org/plosone/s/data-availability#loc-unacceptable-data-access-restrictions.

Reviewers' comments:

Reviewer's Responses to Questions

**Comments to the Author**

1. Is the manuscript technically sound, and do the data support the conclusions?

Reviewer #1: Partly

Reviewer #2: Yes

2. Has the statistical analysis been performed appropriately and rigorously? 

Reviewer #1: No

Reviewer #2: Yes

3. Have the authors made all data underlying the findings in their manuscript fully available?

Reviewer #1: No

Reviewer #2: No

4. Is the manuscript presented in an intelligible fashion and written in standard English?

Reviewer #1: Yes

Reviewer #2: Yes

5. Review Comments to the Author

Reviewer #1: This paper compares the service needs, service utilization, and HIV outcomes of people living with HIV receiving Ryan White services across sex/gender identity. The data are taken from assessments part of the Ryan White program. The authors found that transgender women had higher service needs than cisgender women and men. They also found that transgender women were less likely to use food and nutrition services, and that they were less likely to be durably virally suppressed.

This is a well-presented manuscript that could make an important contribution to our understanding of HIV outcome disparities based on sex/gender. The use of program data (with a large sample) that includes actual measurements of HIV outcomes is strong (compared to, for instance, a study that would be based on a convenience sample or self-reports). Though I see a lot of potential in this manuscript, I have several reservations with the analysis that I believe warrant a major revision before I can recommend for publication.

1. It doesn’t seem like the authors conducted post-hoc tests to assess differences between their three groups of interest (trans women, ciswomen, and cismen) although they report the results as if they did. The chi-square test only assesses whether two variables are significantly associated, but it doesn’t tell which of the subgroups differ significantly from one another. For example, if percentages for group a, b, and c are 16, 20, and 24% respectively, it is possible that group b is not significantly different from either group a or c, and that the only significant difference is between a and c. In this case, we could report that group a was significantly less like to XXX than group c, but not different from group b. To obtain such details, typically a post-hoc z test with p values corrected with the Bonferroni method is used (which is only a matter a checking a few extra boxes for SPSS). The authors should add a column to their tables reporting which group differences were significant (e.g., a<b,c a="" or="">

2. As per guidelines for the journal (which are the standard for most journals), please report test statistics in the tables (i.e., chi-square) and p values to no less than <.001. https://journals.plos.org/plosone/s/submission-guidelines

3. I’m not fully convinced by how the authors defined and operationalized “need.” For instance, “harm reduction need” was defined as “recent” substance use (page 8; also please define “recent”). Is it fair to conclude that anyone who has recently used certain substances has a need for harm reduction? In any case, I think it would be better to report the variable as what it actually measured, that is, substance use. In this case, it would be more accurate to report that X% of participants used substances recently and only Y% of them had accessed harm-reduction services.

4. The definition of “food and nutrition need” also seemed problematic, although the authors recognized that in the Discussion (page 13). I don’t know that it’s accurate to say that everyone under a certain income level has food and nutrition needs. If the authors have a strong rationale for doing so, they should explain it. Otherwise, they may reconsider their operationalization of “need.”

5. The results report on service “need” and utilization, and then on HIV outcomes; however, there is no connection between the two areas (services and HIV outcomes). In the discussion, the authors explain that poorer HIV outcomes among trans women might be due to less service utilization. Why wasn’t service need or utilization included as a potential predictor of HIV outcomes?

6. Looking at HIV outcomes, the authors did multivariable analyses controlling for age, race/ethnicity, and country of birth. However, there is no mention of bivariate tests to determine which control variables to include. It would be appropriate to report what tests were done to establish why these three control variables were selected. The results of bivariate analyses could be reported as supplemental material, if space is a concern.

7. In table 5, because the confidence interval for durable viral suppression includes 1, I don’t think it’s appropriate to report as statistically significant. It seems like trans women were not significantly different from cis men with regards to durable viral suppression. However, cis women were significantly more likely to present durable viral suppression than cis men (probably compared to trans women as well). Why were cis men chosen as the reference group? In any case, this result and associated conclusions should be revisited.

8. On page 7, line 143, “Employment status was categorized as “employed” (for full-time or part-time employment); “unemployed” (for unemployed or unpaid volunteer/peer worker status); and “out of workforce” (for student, retired or homemaker status).” In which category was put a student who is also employed?

9. Page 12, line 223, the authors mention the “ART use measures”. Which ones of the variables are referred to as being about ART use? ART prescription and viral suppression don’t clearly measure use of medication (which sounds more like adherence).

10. Were there participants who did not fit in the three sex/gender categories examined, for instance transgender men or nonbinary individuals? If so, please explain the decision to exclude from the analysis and how many were excluded.</b,c>

Reviewer #2: In summary, this is a very well-conducted paper on HIV disparities among transgender women. The study is very well done and easy to read and follow. Only a very few minor suggestions below. Really, awesome job!

1. In the introduction, I found myself wondering whether there were also disparities by race/ethnicity? This isn’t key to the article so don’t feel the need to add it, but a sentence may just help set the context more.

2. Are participants compensated for their time at all? I don’t think so since these are services under RWPA activities but it may make sense to state this clearly. Up to the authors.

3. Table 1 is missing p-values or, at a minimum, any indication of significance. Please add these.

4. Second and third paragraphs of results are missing any mention of these results being from Tables 2 and 3.

6. PLOS authors have the option to publish the peer review history of their article (what does this mean?). If published, this will include your full peer review and any attached files.

Reviewer #1: No

Reviewer #2: No

---

## [Author Response · Author response to Decision Letter 0]

22 Jan 2021

January 22, 2021

Dear Dr. Morgan, 

My co-authors and I are pleased at the opportunity to revise and resubmit our manuscript entitled ‘Service utilization and HIV outcomes among transgender women receiving Ryan White Part A services in New York City’ (PONE-S-20-38897). 

We are grateful for your constructive comments and suggestions on our manuscript. We have carefully considered all the comments, and respond to each critical point below, in blue, noting where revisions have been made to the manuscript. We explained why no participant consents were required for this retrospective analysis, updated Tables 2 and 3 to reflect the p-values according to PLOS ONE’s style requirements, and included the new results using cisgender women instead of cisgender men as the reference group in Table 5. 

We have uploaded both a clean and a tracked version of the revised manuscript.

Responses to reviewers’ comments

1. Please provide additional details regarding participant consent. In the ethics statement in the Methods and online submission information, please ensure that you have specified (1) whether consent was informed and (2) what type you obtained (for instance, written or verbal, and if verbal, how it was documented and witnessed). If your study included minors, state whether you obtained consent from parents or guardians. If the need for consent was waived by the ethics committee, please include this information.

Response: No participant consents were required for this retrospective analysis, which utilized secondary data that have been reported to the NYC Health Department by our Ryan White Part A (RWPA)-funded service provider agencies and by HIV medical care providers and laboratories, as mandated by NYS laws. Both the RWPA and surveillance data sets that were used were fully de-identified prior to analysis. Only designated staff members who have undergone confidentiality training have access to these data sets. All designated staff members are expected to adhere to the Department of Health data security and confidentiality protocols. This analysis plan was reviewed by the NYC Department of Health IRB and was categorized as public health surveillance, not human subjects research. As a result, no informed consent requirement applies. 

Response: Due to legal restrictions (under New York Public Health Law Article 21, Title III) and the confidential nature of HIV surveillance data in New York, public health authorities in New York City cannot release individual-level data on reported HIV cases for purposes other than ensuring appropriate HIV care. This restriction applies even to de-identified patient-level datasets. However, NYC DOHMH staff are available to assist external researchers who may have further specific data questions or uses. An email can be sent to hivreport@health.nyc.gov with questions or requests for additional information, which will be answered promptly by NYC DOHMH staff.

Review Comments to the Author

1. It doesn’t seem like the authors conducted post-hoc tests to assess differences between their three groups of interest (trans women, ciswomen, and cismen) although they report the results as if they did. The chi-square test only assesses whether two variables are significantly associated, but it doesn’t tell which of the subgroups differ significantly from one another. For example, if percentages for group a, b, and c are 16, 20, and 24% respectively, it is possible that group b is not significantly different from either group a or c, and that the only significant difference is between a and c. In this case, we could report that group a was significantly less like to XXX than group c, but not different from group b. To obtain such details, typically a post-hoc z test with p values corrected with the Bonferroni method is used (which is only a matter a checking a few extra boxes for SPSS). The authors should add a column to their tables reporting which group differences were significant (e.g., a)

Response: Recently, experts have recommended moving away from statistical testing to estimation (Amrhein V, Greenland S, McShane B. Retire statistical significance. Nature 2019;567:305-307.; Greenland S, Senn SJ, Rothman KJ, et al. Statistical tests, P values, confidence intervals and power: a guide to misinterpretations. Eur J Epidemiol 2016;31:337-350.; Wasserstein R, Lazar NA. The ASA statement on p-values: context, process, and purpose. Am Stat 2016;70:129-133.). Following their recommendations, we draw our conclusions of differences across groups based on the effect size, i.e., prevalence ratios, and confidence intervals, not the overall p-values or the p-values from pairwise comparisons with Bonferroni adjustment. We present the overall p-value as a statistical summary of the compatibility between the observed data and what we would expect to see if there were no differences.

2. As per guidelines for the journal (which are the standard for most journals), please report test statistics in the tables (i.e., chi-square) and p values to no less than <.001. https://journals.plos.org/plosone/s/submission-guidelines

Response: In the revised manuscript, we have updated Tables 2 and 3 according to the PLOS ONE guidelines, which state the following on p-values: “Report exact p-values for all values greater than or equal to 0.001. P-values less than 0.001 may be expressed as p < 0.001, or as exponentials in studies of genetic associations.” 

3. I’m not fully convinced by how the authors defined and operationalized “need.” For instance, “harm reduction need” was defined as “recent” substance use (page 8; also please define “recent”). Is it fair to conclude that anyone who has recently used certain substances has a need for harm reduction? In any case, I think it would be better to report the variable as what it actually measured, that is, substance use. In this case, it would be more accurate to report that X% of participants used substances recently and only Y% of them had accessed harm-reduction services.

Response: The definition that we applied matches the eligibility criteria for RWPA Harm Reduction programs. We recognize that it is not a perfect measure for assessing individual need for substance use-related services. Given that we cannot obtain a more nuanced assessment of problem substance use from routine RWPA provider reporting, we treated ‘hard drug’ use (defined as use of cocaine/crack, heroin, crystal meth, or prescription drugs to get high in the past 3 months) as a proxy for need, leveraging existing data. We have replaced the term ‘recent’ in the revised manuscript, in favor of specifying the 3-month timeframe (line 166, page 8). We limited the measure of need to focus on hard drugs because of their association with dependency or other harms to health and safety. We did not include all substance use (e.g., marijuana use or alcohol use), out of recognition that some substances may be used recreationally with moderation, such that reporting some use does not indicate a need for services to reduce use or reduce the harms from use. Furthermore, based on prior analyses in NYC RWPA programs, hard drug use has been found to be associated with worse health outcomes (Feldman MB, Kepler KL, Irvine MK, Thomas JA. Associations between drug use patterns and viral load suppression among HIV-positive individuals who use support services in New York City. Drug and Alcohol Dependence. 2019; 197:15-21.). As mentioned in the paper, there are some limitations related to the substance use data available for analysis. In the RWPA program, substance use is assessed via self-report, and therefore the data can be subject to social desirability bias, which would result in the under-ascertainment of drug use. 

4. The definition of “food and nutrition need” also seemed problematic, although the authors recognized that in the Discussion (page 13). I don’t know that it’s accurate to say that everyone under a certain income level has food and nutrition needs. If the authors have a strong rationale for doing so, they should explain it. Otherwise, they may reconsider their operationalization of “need.”

Response: For clients who were not assessed for food and nutrition need, we based the definition on the national cutoff for SNAP eligibility (130% of the Federal Poverty Level), due to the prevalence of food insufficiency among low-income individuals affected by HIV (Pellowski JA, Kalichman SC, Matthews KA, et al. A pandemic of the poor: social disadvantage and the U.S. HIV epidemic. Am Psychol. 2013;68:197–209.). While we agree that not all clients with that income level have food and nutrition service need, given the cost of living in NYC, people at or below that income level in NYC are likely to experience food insufficiency. In addition, households with incomes below 185% of the poverty threshold (34.3% of all U.S. households) have been found to have higher rates of food insufficiency than the national average, regardless of HIV status (Coleman-Jensen A, Nord M, Singh A. Household Food Security in the United States in 2012, ERR-155, Washington, DC: U.S. Department of Agriculture, Economic Research Service; 2013.).

5. The results report on service “need” and utilization, and then on HIV outcomes; however, there is no connection between the two areas (services and HIV outcomes). In the discussion, the authors explain that poorer HIV outcomes among trans women might be due to less service utilization. Why wasn’t service need or utilization included as a potential predictor of HIV outcomes?

Response: We cannot make that conclusion based on the analyses that were performed for this manuscript. It would have required analyses beyond the scope of this paper, such as looking at doses of services received and change in the outcomes over time (following initiation of services), in order to understand the relationship between service utilization and health outcomes. Also, the RWPA services on which we have data are not the only relevant services a client may be receiving. For example, services funded by Medicaid, Medicare, the Veteran's Administration, and other funding streams outside of RWPA were not available for our analysis. Therefore, we didn’t look at service need or utilization as a potential predictor of HIV outcomes. However, in the discussion, we suggested a potential link between service utilization and outcomes based on the literature, and not based on the specific analyses presented in the paper. 

6. Looking at HIV outcomes, the authors did multivariable analyses controlling for age, race/ethnicity, and country of birth. However, there is no mention of bivariate tests to determine which control variables to include. It would be appropriate to report what tests were done to establish why these three control variables were selected. The results of bivariate analyses could be reported as supplemental material, if space is a concern.

Response: We selected age, race/ethnicity, and country of birth as covariates using a directed acyclic graph, by following guidance for authors from editors of respiratory, sleep, and critical care journals (Lederer D, Bell S, Branson R, et al. Control of confounding and reporting of results in causal inference studies. Guidance for authors from editors of respiratory, sleep, and critical journals. Ann Am Thorac Soc 2019;16:22-28.). A previous study has found that bivariable analysis may not be appropriate to screen risk factors to be included in multivariable analysis (Sun G, Shook TL, Kay GL. Inappropriate use of bivariable analysis to screen risk factors for use in multivariable analysis. J Clin Epidemiol 1996;49:907-916.).

7. In table 5, because the confidence interval for durable viral suppression includes 1, I don’t think it’s appropriate to report as statistically significant. It seems like trans women were not significantly different from cis men with regards to durable viral suppression. However, cis women were significantly more likely to present durable viral suppression than cis men (probably compared to trans women as well). Why were cis men chosen as the reference group? In any case, this result and associated conclusions should be revisited.

Response: We agree that cisgender women should be the reference group. We have included the new results using cisgender women as the reference group in Table 5 and updated the manuscript where applicable. The new Table 5 shows that, compared with cisgender women, transgender women were 20% (aPR = 0.80; 95% CI: 0.69, 0.94) less likely to have durable viral suppression, and the confidence interval does not contain 1.

8. On page 7, line 143, “Employment status was categorized as “employed” (for full-time or part-time employment); “unemployed” (for unemployed or unpaid volunteer/peer worker status); and “out of workforce” (for student, retired or homemaker status).” In which category was put a student who is also employed?

Response: All clients reporting any form of paid employment, including students, were categorized as employed. We have clarified the language in the revised manuscript (line 143, page 7) to indicate that the levels of employment, collected during assessment, are mutually exclusive.

9. Page 12, line 223, the authors mention the “ART use measures”. Which ones of the variables are referred to as being about ART use? ART prescription and viral suppression don’t clearly measure use of medication (which sounds more like adherence).

Response: We are referring to ART prescription, derived from the question ‘Is the client currently prescribed ART?’. We have changed the wording/labeling in the revised paper accordingly to match to what is being collected from RWPA enrollees.

10. Were there participants who did not fit in the three sex/gender categories examined, for instance transgender men or nonbinary individuals? If so, please explain the decision to exclude from the analysis and how many were excluded.

Response: We used two variables, sex assigned at birth and gender, to define the three groups. There were 31 transgender men and 3 non-binary individuals who met the overall eligibility criteria but were excluded from the analysis. Transgender men and non-binary individuals were not included as separate groups because of their small numbers in the client cohort available for analysis. 

Reviewer #2: 

1. In the introduction, I found myself wondering whether there were also disparities by race/ethnicity? This isn’t key to the article so don’t feel the need to add it, but a sentence may just help set the context more.

Response: This is a great point. However, we did not expect to detect conventional disparities by race/ethnicity, mainly because the overwhelming majority of clients in the NYC RWPA program and in this particular cohort identify as Black and Latinx. For example, overall, 13,196 RWPA clients (88%) and 414 transgender women (91%) included in this analysis were Black or Latinx. 

2. Are participants compensated for their time at all? I don’t think so since these are services under RWPA activities but it may make sense to state this clearly. Up to the authors.

Response: No, they are not compensated. There was no time requirement for inclusion in the analysis, since we used existing data sets produced as part of routine reporting on HIV services and HIV laboratory monitoring.

3. Table 1 is missing p-values or, at a minimum, any indication of significance. Please add these.

Response: We understand p-values as measures for inferential purposes, not descriptive ones. Therefore, we did not include p-values in Table 1, but did include them in other tables when we compared outcomes across groups. (Turkiewicz A, Luta G, Hughes HV. Statistical mistakes and how to avoid them -- lessons learned from the reproducibility crisis. Osteoarthritis Cartilage 2018;26:1409e1411.). We will add the p-values to Table 1, if it is a PLOS ONE requirement.

4. Second and third paragraphs of results are missing any mention of these results being from Tables 2 and 3.

Response: In the revised manuscript, we have indicated in those paragraphs which results are from Tables 2 and 3.

Sincerely,

Jacinthe Thomas (for the authors)

Senior Research Analyst, Care & Treatment Research & Evaluation Unit

Bureau of HIV, New York City Department of Health & Mental Hygiene

Phone: (917) 648-2898

E-mail: jthomas1@health.nyc.gov

---

## [Decision Letter · Decision Letter 1]

5 May 2021

PONE-D-20-32483R1

Service utilization and HIV outcomes among transgender women receiving Ryan White Part A services in New York City

PLOS ONE

Dear Dr Thomas,

Thank you for submitting your manuscript to PLOS ONE. After careful consideration, we feel that it has merit but does not fully meet PLOS ONE’s publication criteria as it currently stands. Therefore, we invite you to submit a revised version of the manuscript that addresses the points raised during the review process.

The authors should address and incorporate #2 and 3 from Reviewer 1.

We look forward to receiving your revised manuscript.

Kind regards,

Professor Kwasi Torpey, MD PhD MPH

Academic Editor

PLOS ONE

Journal Requirements:

Additional Editor Comments (if provided):

The revised manuscript titled "Service utilization and HIV outcomes among transgender women receiving Ryan White

Part A services in New York City" was reviewed in response to reviewers' comments. Most of the comments were satisfactorily addressed. However, there were a few comments that the authors provide a rationale citing relevant literature to support their approach. However, I strongly recommend the authors to address #2 and 3 raised by Reviewer 1 pasted below

2. In their response, the authors mention their justification for the covariates they have included in their regression models. This explanation should be included in the manuscript.

3. Though justified, the exclusion of a small number of transgender men and nonbinary individuals from the analytic sample should be mentioned in the manuscript.

Reviewers' comments:

Reviewer's Responses to Questions

**Comments to the Author**

1. If the authors have adequately addressed your comments raised in a previous round of review and you feel that this manuscript is now acceptable for publication, you may indicate that here to bypass the “Comments to the Author” section, enter your conflict of interest statement in the “Confidential to Editor” section, and submit your "Accept" recommendation.

Reviewer #1: (No Response)

Reviewer #2: All comments have been addressed

2. Is the manuscript technically sound, and do the data support the conclusions?

Reviewer #1: Partly

Reviewer #2: Yes

3. Has the statistical analysis been performed appropriately and rigorously? 

Reviewer #1: No

Reviewer #2: Yes

4. Have the authors made all data underlying the findings in their manuscript fully available?

Reviewer #1: (No Response)

Reviewer #2: Yes

5. Is the manuscript presented in an intelligible fashion and written in standard English?

Reviewer #1: Yes

Reviewer #2: Yes

6. Review Comments to the Author

Reviewer #1: The authors did a minor revision of their manuscript; some of my concerns from the initial submission remain.

1. To my comment asking for subgroup comparisons, the authors offered resources warning against common misuse of statistical significance. I carefully looked through these resources and do not believe the authors of this manuscript have followed their recommendations. In their responses, the authors say these experts recommend “moving away from statistical testing,” but I understand these articles to be warning us against misinterpretations of statistical significance.

For example, at lines 213–216 of their manuscript, the authors state “Compared to cisgender women and cisgender men, significantly higher proportions of transgender women had an apparent need for support in the areas of housing (52% versus 24% and 35%, respectively), mental health (24% versus 22% and 20%, respectively), or harm reduction (23% versus 12% and 18%, respectively) (Table 3).” Here, the authors rely on the statistically significant p value to report that transgender women had “significantly” higher need in the area of mental health. However, if the authors did not simply rely p values to make conclusions—as they indicate in their responses—they would remark that the proportion of transgender women and cisgender women with mental health needs were not highly different (24% and 22%) instead of relying on the statistical test to claim a “significant” difference. What is more, as I pointed in my initial review, there is no evidence that these two proportions are statistically different because there were no subgroup comparisons. As such, the authors seem to be drawing conclusions based on p values (and incorrect ones). Although I provide only one example, most of the results of the paper are reported similarly and the abstract and discussion rely on subgroup comparisons that are not well supported.

If the authors wish to move away from statistical testing, they should not rely on those tests to make statements of difference. However, if I understood the resources provided correctly, statistical tests should still be done (and done correctly), but researchers should provide more nuanced discussions and conclusions of the results that do not simply make a dichotomy between what is significant and not.

2. In their response, the authors mention their justification for the covariates they have included in their regression models. This explanation should be included in the manuscript.

3. Though justified, the exclusion of a small number of transgender men and nonbinary individuals from the analytic sample should be mentioned in the manuscript.

4. I believe the author guidelines also require including test statistics (chi-square value), not only p values (for Tables 2 and 3).

Reviewer #2: I have no additional comments, all of my previous concerns have now been adequately addressed. Thank you!

7. PLOS authors have the option to publish the peer review history of their article (what does this mean?). If published, this will include your full peer review and any attached files.

Reviewer #1: No

Reviewer #2: No

---

## [Author Response · Author response to Decision Letter 1]

28 May 2021

May 28, 2021

Dear Dr. Torpey, 

We appreciate your consideration of our revised manuscript entitled ‘Service utilization and HIV outcomes among transgender women receiving Ryan White Part A services in New York City’ (PONE-S-20-38897). Thank you for extending the resubmission deadline.

 We are grateful for the opportunity to fully address one major concern from the prior review on the test that was used to assess differences between the groups. We have conducted the suggested statistical tests and updated Tables 2, 3, and 4 to add chi-square values and p-values from pairwise comparisons, and split Table 3 in two to avoid having too much information in one table. We have responded to each of the comments below, in blue, noting where revisions have been made to the manuscript.

We are encouraged to see that Reviewer #2 felt the revised version of the

manuscript had addressed all concerns included in the prior review. We have uploaded both a clean and a tracked version of the revised manuscript.

Responses to comments from Reviewer #1

1. To my comment asking for subgroup comparisons, the authors offered resources warning against common misuse of statistical significance. I carefully looked through these resources and do not believe the authors of this manuscript have followed their recommendations. In their responses, the authors say these experts recommend “moving away from statistical testing,” but I understand these articles to be warning us against misinterpretations of statistical significance.

For example, at lines 213–216 of their manuscript, the authors state “Compared to cisgender women and cisgender men, significantly higher proportions of transgender women had an apparent need for support in the areas of housing (52% versus 24% and 35%, respectively), mental health (24% versus 22% and 20%, respectively), or harm reduction (23% versus 12% and 18%, respectively) (Table 3).” Here, the authors rely on the statistically significant p value to report that transgender women had “significantly” higher need in the area of mental health. However, if the authors did not simply rely p values to make conclusions—as they indicate in their responses—they would remark that the proportion of transgender women and cisgender women with mental health needs were not highly different (24% and 22%) instead of relying on the statistical test to claim a “significant” difference. What is more, as I pointed in my initial review, there is no evidence that these two proportions are statistically different because there were no subgroup comparisons. As such, the authors seem to be drawing conclusions based on p values (and incorrect ones). Although I provide only one example, most of the results of the paper are reported similarly and the abstract and discussion rely on subgroup comparisons that are not well supported.

If the authors wish to move away from statistical testing, they should not rely on those tests to make statements of difference. However, if I understood the resources provided correctly, statistical tests should still be done (and done correctly), but researchers should provide more nuanced discussions and conclusions of the results that do not simply make a dichotomy between what is significant and not.

Response: We agree with the reviewer’s comment. In order to mention significant differences between the subgroups, we must rely on post-hoc tests and p-values from pairwise comparisons for such an assessment. Therefore, we have included the chi-square values and p-values in Table 2, 3, 4, and 5 and updated the manuscript where applicable. 

2. In their response, the authors mention their justification for the covariates they have included in their regression models. This explanation should be included in the manuscript.

Response: In the revised manuscript, we have added a sentence to indicate how we decided on those three covariates.

3. Though justified, the exclusion of a small number of transgender men and nonbinary individuals from the analytic sample should be mentioned in the manuscript.

Response: In the revised manuscript, we have indicated that transgender men and nonbinary individuals were excluded from the analysis due to their small numbers.

4. I believe the author guidelines also require including test statistics (chi-square value), not only p values (for Tables 2 and 3).

Response: In the revised manuscript, we have updated Tables 2, 3, and 4 and added both the chi-square values and p-values from pairwise comparisons.

5. Thank you for including your ethics statement in your Response to Reviewers: "No participant consents were required for this retrospective analysis, which utilized secondary data that have been reported to the NYC Health Department by our Ryan White Part A (RWPA)-funded service provider agencies and by HIV medical care providers and laboratories, as mandated by NYS laws. Both the RWPA and surveillance data sets that were used were fully de-identified prior to analysis. Only designated staff members who have undergone confidentiality training have access to these data sets. All designated staff members are expected to adhere to the Department of Health data security and confidentiality protocols. This analysis plan was reviewed by the NYC Department of Health IRB and was categorized as public health surveillance, not human subjects research. As a result, no informed consent requirement applies.".

To help ensure that the wording of your manuscript is suitable for publication, would you please also add this statement at the beginning of the Methods section of your manuscript file.

Response: To comply with the submission guidelines, we have expanded the Ethics language into a full paragraph in the revised manuscript and added further subheadings in the Methods section to ensure that the focus of each Methods sub-section was clearly introduced.

Sincerely,

Jacinthe Thomas (for the authors)

Senior Research Analyst, Care & Treatment Research & Evaluation Unit

Bureau of HIV, New York City Department of Health & Mental Hygiene

Phone: (917) 648-2898

E-mail: jthomas1@health.nyc.gov

---

## [Editor Report · Decision Letter 2]

7 Jun 2021

Service utilization and HIV outcomes among transgender women receiving Ryan White Part A services in New York City

PONE-D-20-32483R2

Dear Dr. Thomas,

We’re pleased to inform you that your manuscript has been judged scientifically suitable for publication and will be formally accepted for publication once it meets all outstanding technical requirements.

Kind regards,

Professor Kwasi Torpey, MD PhD MPH

Academic Editor

PLOS ONE

Additional Editor Comments (optional):

Outstanding issues addressed
---

## [Editor Report · Acceptance letter]

23 Jun 2021

PONE-D-20-32483R2 

Service utilization and HIV outcomes among transgender women receiving Ryan White Part A services in New York City 

Dear Dr. Thomas:

I'm pleased to inform you that your manuscript has been deemed suitable for publication in PLOS ONE. Congratulations! Your manuscript is now with our production department. 

Kind regards, 

on behalf of

Professor Kwasi Torpey 

Academic Editor

PLOS ONE